# Association Between Zinc Status and Insulin Resistance/Sensitivity Check Indexes in Gestational Diabetes Mellitus

**DOI:** 10.3390/ijms252212193

**Published:** 2024-11-13

**Authors:** Mariana P. Genova, Irena Ivanova, Emilia Naseva, Bisera Atanasova

**Affiliations:** 1Department of Clinical Laboratory, Medical University of Sofia, Alexander University Hospital, 1431 Sofia, Bulgaria; mariana8sofia@yahoo.com (M.P.G.); bissera.atanassova@gmail.com (B.A.); 2Department of Clinical Laboratory, UH St. Ivan Rilski, 1606 Sofia, Bulgaria; irena.dimitrova@gmail.com; 3Department of Health Management and Health Economics, Faculty of Public Health “Prof. Tsekomir Vodenicharov, MD, DSc”, Medical University of Sofia, 1524 Sofia, Bulgaria; 4Medical Faculty, Sofia University St. Kliment Ohridski, 1504 Sofia, Bulgaria

**Keywords:** gestational diabetes mellitus, zinc status, Glu/Zn, Ins/Zn and HOMA-IR/Zn indexes

## Abstract

Gestational diabetes mellitus (GDM) is considered the most common metabolic disorder of the pregnancy period. It is characterized by pancreatic beta-cell dysfunction in the setting of chronic insulin resistance. Zinc is a nutrient involved in numerous metabolic processes and shows a relationship with glycometabolic disorders and GDM. The latest data have demonstrated the association of zinc with insulin sensitivity and resistance. The exact role of zinc in the connection with indexes of insulin resistance and insulin sensitivity is still not fully clarified. The aim of the study is to analyze the newly calculated indexes Glu/Zn, Ins/Zn, and HOMA-IR/Zn as surrogate markers to explore the correlation between serum zinc status and some indexes of insulin sensitivity and insulin resistance. The possible role of these indexes as markers of insulin resistance in pregnant women was analyzed too. An ROC analysis demonstrated that HOMA-IR/Zn with AUC 0.989, *p* < 0.001 (95% CI 0.967–1.000) and Ins/Zn with AUC 0.947, *p* < 0.001 (95% CI 0.889–1.000) in the GDM group, and only HOMA-IR/Zn index with AUC 0.953, *p* < 0.001 (95% CI 0.877–1.000) in healthy pregnant women, have good power as markers of insulin resistance in both groups. We speculate that these new ratios could be suitable for the assessment of pregnant women at high risk of insulin resistance development and, probably, for the evaluation of the specific pathophysiologic characteristics of women with GDM.

## 1. Introduction

Gestational diabetes is defined as a glucose intolerance of variable severity with onset during pregnancy [1]. It is a common disease during pregnancy and reflects a set of endocrine complications arising from the failure in the adaptive physiologic capability of the pancreas. GDM is considered the most common metabolic disorder of the pregnancy period with negative short- and long-term effects on both maternal and fetal health [2,3]. According to the International Association of Diabetes and Pregnancy Study Group (IADPSG) criteria, the global prevalence of GDM was 14.0% (95% confidence interval, CI: 13.97–14.04%) in 2021 and regional prevalence ranges from 7.1–27.6% [4]. Gestational diabetes mellitus is a result of pancreatic beta-cell dysfunction in the setting of chronic insulin resistance during pregnancy and thus both beta-cell impairment and tissue insulin resistance represent the main mechanisms for GDM development [5].

Trace elements support numerous biochemical reactions [6] including those related to insulin and glucose metabolism. Disturbances in the levels of essential elements may lead to the development of GDM by causing disorders in insulin sensitivity, insulin resistance and glucose intolerance [7,8].

Zinc is an essential micronutrient involved in many cellular processes such as protein and DNA synthesis, nucleic acid metabolism, gene transcription [9], cellular replication and differentiation, and hormone regulation [10]. Zinc mediates its metabolic actions via the action of several zinc transporters that regulate zinc homeostasis and control its cellular compartmentalization [11]. This determines that intracellular zinc homeostasis as a whole, and in particular the intracellular presence of zinc-binding proteins like metallothioneins (MTs), are important for normal cell function in various tissues [12].

Zinc plays an important role in the endocrine system. It has been previously shown that there is an association between zinc and both insulin sensitivity and resistance and that zinc has the ability to regulate insulin receptors and directly modulate insulin activity and glucose homeostasis. [13]. Several studies have investigated the relationship of zinc with Type 2 Diabetes Mellitus (T2DM) and glycometabolic disorders, but the conclusions are inconsistent [14,15]. A meta-analysis conducted by Yang et al. detected that zinc supplementation could significantly improve fasting plasma glucose (FPG) and improve HOMA-IR, glycated hemoglobin, and 2 h-plasma glucose [16].

Zinc plays a crucial role in the healthy development of the embryo during pregnancy. Adequate maternal zinc status is very important for the transfer of zinc to the fetus [17]. In the first weeks of pregnancy, zinc is required for cell multiplication and differentiation during the processes of fetal organ formation. The deficiency of zinc, mainly in later pregnancy, adversely impacts normal neuronal replication, migration, synaptogenesis, and gene expressions [18].

There has been recent interest in the complex relationship between trace elements and GDM. Evidence showed the associations between serum zinc, glucose levels, and indexes of insulin resistance and insulin sensitivity during GDM pregnancy using zinc supplements [19,20] and revealed the beneficial effect of zinc on FPG and markers of insulin resistance.

There are few studies examining the potential association between zinc levels and GDM to date. Mishu et al. [21] reported that the maternal serum zinc concentration during pregnancy is associated with GDM incidence. A meta-analysis and systematic review by Fan et al. [22] found that the measurement of zinc and other trace elements during the second trimester of the gestation period helps to distinguish high-risk cases for GDM.

The newly calculated indexes Glu/Zn, Ins/Zn, and HOMA-IR/Zn are presented as surrogate markers of the relationship between serum zinc status, the secretory function of pancreatic beta cells, and the homeostasis model assessment of insulin resistance (HOMA-IR) [23].

The aim of the study is to analyze the correlation between the aforementioned indexes of zinc status, indexes of insulin resistance, and insulin sensitivity in pregnant women with GDM and normal glucose tolerance (NGT) and to assess their possible role as markers of insulin resistance in pregnant women.

## 2. Results

### 2.1. Characteristics and Comparison of Pregnant Women with and Without GDM

One hundred and eight pregnant women were included in the study. Diagnosis of GDM was made based on an abnormal 75 g oral glucose tolerance test at 23–28 gestational weeks (gw). Fifty-four pregnant women were diagnosed with GDM and 54 did not have GDM. The general characteristics and comparison of variables for the two groups included in the study are presented in Table 1.

Compared to women without GDM, pregnant women with GDM were statistically significantly more likely (*p* < 0.05) to have a family history of diabetes, higher pre-pregnancy BMI at the GDM diagnosis, and higher concentrations of FPG and glucose levels post-1 h and 2h OGTT. Pregnant women with GDM presented significantly higher FSI, HOMA-IR; HOMA-B, HOMA-S, and QUICKI indexes (all *p* < 0.05). No relevant differences were shown in age, gestational age between groups, and zinc level.

The levels of Glu/Zn, Ins/Zn, and HOMA-IR/Zn were found to be significantly higher for the GDM group in comparison to the NGT group (*p* < 0.05) as shown in Table 2.

The correlations between Glu/Zn, HOMA-IR/Zn, Ins/Zn indexes, and selected variables of insulin resistance and insulin sensitivity in both studied groups of pregnant women are presented in Table 3 and Table 4.

In the NGT group, there was a weak negative significant correlation between Glu/Zn and HOMA-S for the GDM cohort. There was a weak negative correlation between Glu/Zn and QUICKI in both groups but with no significance for GDM.

The correlation analyses of HOMA-IR/Zn and Ins/Zn with selected markers of insulin resistance and sensitivity are presented in Figure 1 and Figure 2 separately.

### 2.2. Assessment of the Usefulness of the Examined Indexes Glu/Zn, HOMA-IR/Zn, Ins/Zn as Markers of Insulin Resistance Using ROC Curves

An ROC analysis was carried out to define the potential of Glu/Zn, HOMA-IR/Zn, and Ins/Zn as biomarkers of insulin resistance among pregnant women. This study has shown that HOMA-IR/Zn with AUC 0.989, *p* < 0.001 (95% CI 0.967–1.000) and Ins/Zn with AUC 0.947, *p* < 0.001 (95% CI 0.889–1.000) were the only indexes that were statistically significant biomarkers of insulin resistance in the GDM group (Figure 3). Additionally, the differences in AUC values between the two indexes were insignificant (*p* = 0.4) according to the comparison model suggested by DeLong [24].

An ROC curve of examined indexes in the NGT group revealed that only the HOMA-IR/Zn index with AUC 0.953, *p* < 0.001 (95% CI 0.877–1.000) was a statistically significant biomarker of insulin resistance in the NGT group (Figure 4). However, the diagnostic potential of the Glu/Zn index in both groups of pregnant women was not statistically significant (*p* > 0.05).

The optimal cutoff for HOMA-IR/Zn was >0.22 with 93.2% specificity and 100% sensitivity; for Ins/Zn a cutoff of >0.9 yielded 81.8% specificity and 100% sensitivity, respectively, in the GDM group. No threshold of HOMA-IR/Zn was found in the NGT group.

## 3. Discussion

It is now well known that trace elements may play specific roles in the pathogenesis and progression of diabetes. An increasing number of animal and human studies have demonstrated that several essential elements, including zinc (Zn), iron (Fe), manganese (Mg), copper (Cu), and calcium (Ca), can affect glucose metabolism and insulin sensitivity with downstream effects on hyperglycemia and hence GDM [25,26].

In the current study, the values of serum zinc in GDM pregnant women were not found to be significantly different when compared to the NGT group (*p* = 0.872). Earlier work by Hamdan et al. [27] and Wilson et al. [28] demonstrated no significant differences in serum zinc levels between pregnant women with GDM and NGT in late pregnancy. Our previous study did not identify statistically significant differences between healthy pregnant women and those with GDM in the levels of plasma and hemolyzate zinc [29]. Both pregnant groups (second trimester of pregnancy) had higher levels of intracellular erythrocyte zinc in comparison to non-pregnant [29]. Various studies have examined serum levels of zinc in GDM pregnant women with inconsistent results. A study by Deng et al. has shown that serum zinc concentrations in GDM pregnant were markedly higher than in the non-GDM group [30]. The authors pointed out that some [21,31] but not all [32] cross-sectional studies were in agreement with their results, in that patients with T2DM or GDM have lower serum zinc levels due to increased urinary excretion. In a systemic review and meta-analysis by Fun et al., the zinc content demonstrated a significant difference between GDM patients and healthy controls in subgroup analysis in the second trimester. The subgroup analysis stratified based on geographical position, showed that circulating zinc levels amongst the Asia population are lower in the GDM cases without significant difference among the Caucasians and Africans [22]. For the second trimester, Liu et al. reported that the zinc level declined with the progression of pregnancy, suggesting that such reduced zinc content was related to the disproportional elevation of plasma volume and the transfer of zinc from the mother to the fetus [33]. Zinc levels are mainly dependent on dietary intake. Moreover, zinc required for multiple general metabolic functions is physiologically provided not by body stores but by the rate of removing zinc pools. It has no body stores. The zinc level is also a result of the existence of slow and rapid removable zinc pools [34]. Not surprisingly, pregnant women in developed countries normally present better zinc status due to dietary zinc consumption as a result of various national dietary interventions [35]. Liu XB et al. [36] found that serum zinc concentration in healthy pregnant women is higher in comparison to the value reported in previous Chinese studies [37], but slightly lower than the results of American [38], European [39,40], and Korean studies [41]. Other factors, such as infection, inflammation, exercise, stress or trauma, circadian variation, and fasting status could affect serum zinc levels [42].

A healthy pancreas contains relatively high levels of zinc with the predominant concentration within the beta cells, where it is co-released with insulin [43]. Zinc has insulin-mimicking properties, promotes glucose uptake in insulin-dependent tissues [44], and stimulates phosphorylation of the β-subunit of the insulin receptor tyrosine kinase, increasing insulin sensitivity [45]. Furthermore, this trace element is mandatory for induction of the translocation of glucose transporter 4 (GLUT4) [46,47,48]. Apart from this, zinc has the function of modulating the transcription of the insulin receptor gene. The process is mediated by zinc finger proteins containing zinc in their conformation. In particular, zinc finger 407 regulates glucose uptake by improving GLUT4 messenger RNA levels and stimulates its transcription [49].

It is now well known that zinc is involved in blood glucose homeostasis and is associated with insulin resistance, and insulin sensitivity through different mechanisms which are not fully understood and elucidated. This study demonstrated the relationship between zinc status and glycemic biomarkers and found that optimal zinc intake may play a role as a modifiable factor in glycemic control and diabetes prevention. However, this needs to be tested and confirmed in a larger study.

A significant association was observed between serum zinc levels, insulin sensitivity, beta-cell function, and insulin resistance in pediatric patients with no significant association of HOMA parameters in a normoglycemia state [50].

Zinc supplementation significantly reduces the levels of FPG, HOMA-IR, insulin, and glycated hemoglobin as established by umbrella systematic review and meta-analysis [51]. The published results of meta-analyses in the literature are controversial. Some display the beneficial effects of zinc on glycemic biomarkers such as FPG, HOMA-IR, and insulin [16,52], but others report a lack of a significant effect on FPG, glycated hemoglobin, HOMA-IR [53], and insulin [54,55,56].

Zinc supplementation can alter glycemic markers in patients with diabetes and in patients at a high risk of developing diabetes. Zinc supplement intake is associated with diminished blood glucose levels and nonenzymatic glycation, thus raising insulin sensitivity [56].

A comparison between normoglycemia, prediabetes, and diabetes groups indicates that higher serum zinc concentration is associated with lower insulin secretion and lower insulin resistance in prediabetic individuals. The higher serum zinc level is associated with ameliorated pancreatic function in the normoglycemic state [57]. Zinc supplementation has a positive effect on FPG in patients with T2DM [58] and significantly lower FPG, HOMA-IR, glycated hemoglobin, and OGTT-2h in overweight and obese population groups [16]. A notable correlation between FPG and OGTT-2h and serum zinc is indicated in T2DM [59] in contrast to other observations about a significant negative correlation between zinc and FPG [60].

A few studies have examined the potential effects of serum zinc levels on glucometabolic parameters, particularly with respect to pregnant women with GDM. A meta-analysis conducted by Li et al. suggests that zinc supplementation results in significantly decreased FPG, insulin, HOMA-IR, and an increased QUICKI index in GDM [52]. Significant negative associations between plasma zinc concentration and OGTT-1h and OGTT-2 h glucose levels in pregnant women with GDM [61] were established as consistent with growing evidence supporting the relation between hyperglycemia and serum zinc status. In contrast, insufficient evidence was accumulated for improved outcomes by zinc supplementation for mothers and newborns [62].

To date, only limited studies have examined the correlation between markers of insulin resistance and insulin sensitivity (HOMA-IR, HOMA-B, QUCKI index), some glycemic markers (glucose and insulin), and serum zinc levels in GDM, with no statistically significant correlations between zinc and parameters, as mentioned above. Notably, the correlations are not adjusted for confounder factors [63,64]. In addition, plasma zinc is not correlated with insulin and HOMA-IR even after controlling for race and age in healthy early-adolescent girls [65].

In this study, we hypothesized that the relationship between serum zinc level and some glucose metabolic indexes will be more significant if this nutrient element is included in new ratios. We also attempted to ‘normalize’ the generally accepted indexes of be-ta-cells secretory function and markers of insulin resistance to serum zinc and found new surrogate biomarkers, namely Glu/Zn, Ins/ Zn, and HOMA-IR/Zn ratios. Furthermore, we speculated that these ratios may act as useful markers of insulin resistance.

Our results have shown a positive correlation between zinc surrogate indexes HOMA-IR/Zn, Ins/Zn, and HOMA-IR and HOMA-B markers, and a significant negative correlation between these indexes and HOMA-S and the QUICKI index (for all *p* < 0.001) in GDM pregnant women. The outcomes in healthy pregnant women are similar due to the presence of insulin resistance, and slightly lower in comparison to results in the GDM group. 

Grâdinaru et al. suggested these new ratios were helpful in the development of preventive and therapeutic strategies among elderly diabetic patients based on the evaluation of serum zinc levels [23]. In the current study, we found that these new, zinc-containing surrogate biomarkers are involved in glucose homeostasis in pregnant women with and without GDM.

## 4. Materials and Methods

This study is a case–control, single-center study, conducted on pregnant women in the second trimester, between May 2018 and October 2020. All participants were included with signed informed consent. The study is in agreement with the principles expressed in the Declaration of Helsinki [66]. The study was performed with ethical approval from the Medical University Ethics Review Board (Survey code: BK-324/06.03.2018, Ethics approval date: N12/21.03.2018).

The inclusion criteria were as follows: gestational age 23–28 gestational week, singleton pregnancy, and pregnant women older than 18 years. Women were excluded from the survey in cases of pre-pregnancy diabetes; some endocrine, metabolic, and cardiovascular diseases including pre-gestational and gestational hypertension; other cases of chronic diseases (chronic kidney and liver diseases); cases of inflammatory, mental, and infectious diseases; and patients treated with drugs affecting glucose levels. Likewise, women with data for preeclampsia, eclampsia, fetoplacental abnormalities, and a history of previous GDM were excluded. Furthermore, all participants, selected from a cohort of pregnant women, declared a lack of anamnestic data about polycystic ovary syndrome (PCOS). The research study included a group of 54 newly diagnosed pregnant with GDM and a control group of 54 healthy pregnant women after a 2-h OGTT. The mothers’ socio-demographic data, medical history, habits, and behavior factors were recorded in detail, directly employing a structured questionnaire through personal interviews. Variables such as maternal age at recruitment, smoking, and drinking habits, pre-pregnancy height and weight and weight of pregnant women at the diagnosis of GDM, and the gestational week at enrollment were collected by means of a structured questionnaire. The information about all participants in the study was anonymized to preserve their identity.

### 4.1. Diagnosis of GDM

The total group of 108 pregnant women was required to conduct a 2-h 75 g oral glucose tolerance test with fasting. The interpretation of the OGTT results was in accordance with the criteria of the International Association of Diabetes and Pregnancy Study Groups. Typically, GDM was diagnosed if any of the OGTT results fulfilled at least one of the following criteria: FPG ≥ 5.1 mmol/L; OGTT-1h ≥10.0 mmol/L, and OGTT-2h ≥ 8.5 mmol/L (American Diabetes Association 2021) [2].

### 4.2. Sample Collection and Measurement

The blood collection was drawn in the morning, between 7 and 9 am in a fasting state. About 5 mL of fasting venous blood was drawn by experienced phlebotomists with a vacutainer with SST gel and 4 mL venous blood with a vacutainer with K2EDTA from each enrolled woman. After that, all participants were given a 300 mL solution of water containing 75 g of glucose and instructed to drink the solution within 5 minutes for the OGTT. Venous blood samples were collected at 60 and 120 minutes. Fasting venous blood samples were centrifuged for 10 min at 3000 rpm to obtain blood serum. The separated serum samples were transferred into metal-free plastic microcentrifuge tubes and stored at −20 °C for about 7 days until zinc analysis. All blood glucose samples were analyzed using the amperometric method for glucose (Analyzer Biosen C-line, Barleben, Germany). FSI concentration in the serum was analyzed by electrochemiluminescence immunoassay (ECLIA). For this purpose, we used a commercially available kit (Roche Diagnostics, Mannheim, Germany) with the automatic analyzer Elecsys 2010 (Roche Diagnostics, Mannheim, Germany). Serum zinc levels were analyzed by flame atomic absorption spectrophotometry (Perkin Elmer AAnalyst 300, Norwalk, CT, USA); CV% day-to-day variation was 5.2% and bias < 1%.

All laboratory analyses fulfill the demands of the standardization and certification program.

### 4.3. Calculation of Indexes of Interest

Updated HOMA-IR (HOMA2): estimated insulin resistance, pancreatic β cell function (%B), and insulin sensitivity (%S) were obtained using the HOMA2 calculator available at http://www.dtu.ox.ac.uk/homacalculator/index.php (accessed on 8 January 2013) [67].
QUICKI index (quantitative insulin sensitivity check index) = 1/[log10 (FPG [mg/dL]) + log10 (FI [µU/mL])]

Pre-gestational BMI (kg/m^2^); BMI at the GDM diagnosis weight (kg)/height (m)^2^, where the data about weight and height were self-reported during the interview.

The indexes of glycemic control and zinc status were calculated as follows:

Glu/Zn ratio, using the FPG and Zn, expressed as mmol/µmol/L

Ins/Zn ratio, using FSI and Zn, expressed as µU/mL/µmol/L

HOMA-IR/Zn ratio, using HOMA-IR individual value and serum zinc level.

### 4.4. Statistical Analysis

All data were analyzed using IBM Statistical Package for Social Science (SPSS) software for Windows version 26. Results are expressed as frequencies or percentages for qualitative variables and mean (standard deviation) for normally distributed quantitative variables and median and interquartile range (IQR; both 25th and 75th percentile) in not normally distributed ones. For testing the normality of the distribution, the Kolmogorov–Smirnov test (sample size > 50) was used. Comparisons of variables with normal or skewed distribution between the GDM group and the group of healthy pregnant women were performed using the Mann–Whitney or Independent Samples t-test as well as the Pearson chi-squared test. The correlation between Glu/Zn, HOMA-IR/Zn, and Ins/Zn levels and HOMA-IR, HOMA-B, HOMA-S, and QUICKI were analyzed using the Spearman’s rho correlation coefficient. Receiver operating characteristic (ROC) curves and areas under the curve (AUC) were performed to determine the possibility of the indexes Glu/Zn, HOMA-IR/Zn, and Ins/Zn to be used as markers of IR. A *p*-value less than 0.05 is considered statistically significant. Cutoff values are selected at the maximum Youden index.

The sample size was calculated for two means based on the literature for zinc levels among second-trimester GDM and healthy controls [40]. It resulted in 46 pregnant women in each arm of the study, assuming 95% confidence and 80% power. We added some more cases to compensate for possible drop-outs.

## 5. Conclusions

Our pilot study was a single-center study conducted among two study groups. It has several limitations that should be mentioned. The major limitation of the current study is the small number of tested pregnant women. The pandemic of COVID-19 infection was a significant reason for the limitation of the participating number. As a result, subgroup analysis was not performed for this study. Moreover, a small sample size limits the statistical power of the study. Other confounding factors, such as age, pre-pregnancy BMI, BMI at GDM diagnosis, and gestational weeks, were not controlled in our study. Other limitations are no follow-up period and the absence of analysis of proteins involved in zinc metabolism such as zinc-α2 glycoprotein and zinc-induced metallothioneins.

Together with the limitations, we may mention that in this study, to our knowledge, we were the first to investigate these ratios in pregnant with GDM and healthy pregnant women.

Undoubtedly, the newly introduced ratios of Ins/Zn, HOMA-IR/Zn, and Glu/Zn need further research in larger research groups. However, as an initial survey, our results could serve as a useful starting point for future investigations, which may expand the benefit of the presented ratios in clinical practice for the diagnosis of insulin resistance. Additionally, they may be used routinely as numerous other markers to evaluate metabolic impairment and insulin resistance in pregnant women.

While these findings need to be confirmed through further larger and expanded studies, this investigation indicates a new potential role of zinc status and its significant associations with indexes of insulin sensitivity and insulin resistance. The results of this study show that these new surrogate biomarkers can be used as laboratory computational indexes in the diagnosis of insulin resistance in pregnant women. We speculate that these new ratios could be suitable for the assessment of pregnant women at high risk of insulin resistance development and, probably, for the evaluation of some specific pathophysiologic characteristics of women with GDM.

The suggested ratios are not definitive for the diagnosis, but they could be clinically beneficial for pregnancy management, especially in the complicated situation of GDM.

## Figures and Tables

**Figure 1 ijms-25-12193-f001:**
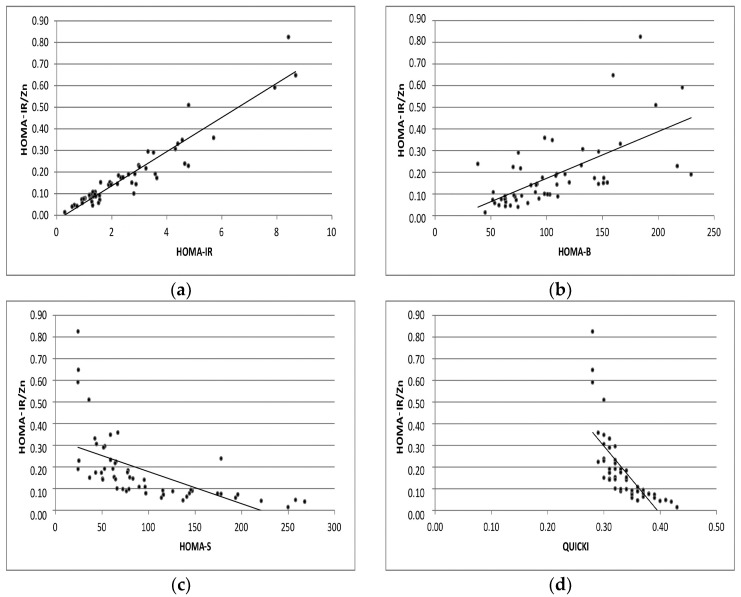
Correlation analyses of HOMA-IR/Zn with levels of HOMA-IR, HOMA-B, HOMA-S, QUICKI indexes (**a**–**d**) in GDM pregnant group.

**Figure 2 ijms-25-12193-f002:**
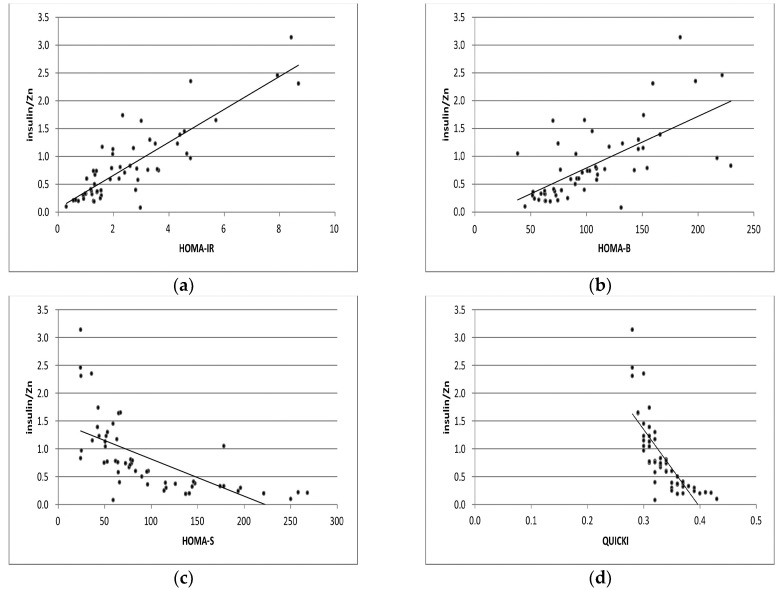
Correlation analyses of Ins/Zn with levels of HOMA-IR, HOMA-B, HOMA-S, and QUICKI indexes (**a**–**d**) in GDM pregnant group, respectively.

**Figure 3 ijms-25-12193-f003:**
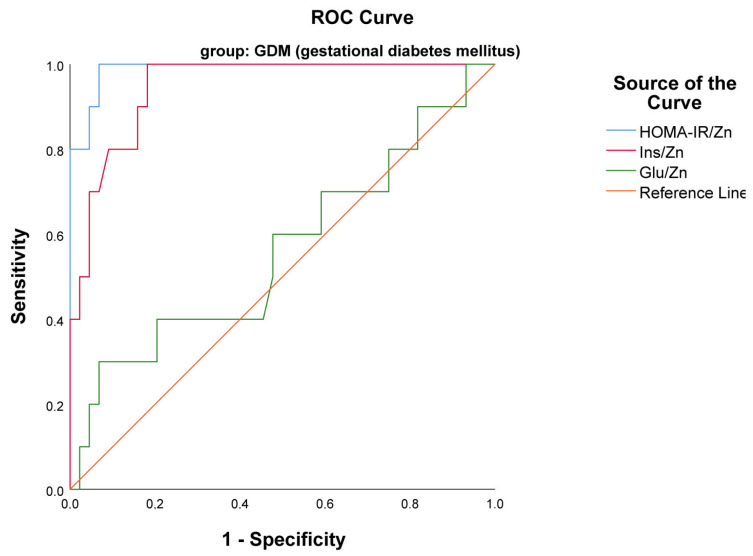
ROC curves presenting the usefulness of the examined indexes HOMA-IR/Zn and Ins/Zn as markers of insulin resistance in GDM group.

**Figure 4 ijms-25-12193-f004:**
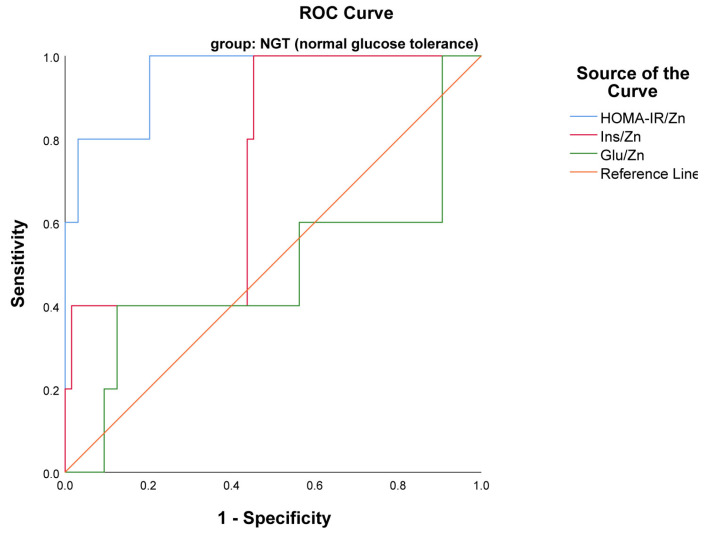
ROC curve presented HOMA-IR/Zn as a marker of insulin resistance in NGT group.

**Table 1 ijms-25-12193-t001:** Comparison of characteristics of the study participants.

Parameters	GDM Pregnant Woman (n = 54)	NGT Pregnant Woman; (n = 54)	*p*-Value
Maternal age (years); mean ± SD	32 ± 5.8	30.4 ± 5.1	0.078 **
Pre-gestational BMI (kg/m^2^); mean ± SD	27.6 ± 7.2	23.3 ± 4.1	0.001 **
BMI at GDM diagnosis (kg/m^2^); mean ± SD	30.4 ± 7.0	26.1 ± 4.2	<0.001 **
Gestational age at blood drawing (weeks); mean ± SD	26.5 ± 3.2	25.9 ± 2.8	0.522 **
Smoking habit; n, %	0	0	-
Drinking habit; n, %	0	0	-
Family history of diabetes; n, %	15(27.7)	6(11.1)	0.016 ***
FPG, mmol/L; median (interquartile range)	5.3(5.1–5.4)	4.4(4.1–4.7)	<0.001 *
OGTT-1h, mmol/L; median (interquartile range)	8.3 ± 2.3	6.7 ± 1.5	<0.001 **
OGTT-2h; mmol/L; median (interquartile range)	6.0(4.7–7.6)	5.0(4.4–6.1)	<0.002 *
FSI µU/mL; median (interquartile range)	10.3(5.4–16.1)	7.3(5.0–10.4)	<0.002 *
HOMA-IR; median (interquartile range)	2.0(1.3–3.4)	0.9(0.6–1.3)	<0.001 *
HOMA-B; median (interquartile range)	94.7(69.4–135)	113.3(84.2–149.5)	0.041 *
HOMA-S; median (interquartile range)	76.8(51.4–138)	126.1(81.7–180.3)	<0.001 *
QUICKI index; median (interquartile range)	0.3(0.3–0.4)	0.4(0.3–0.4)	<0.001 *
Zn, µmol/L; median (interquartile range)	13.7(13–16.8)	15.1(12.4–18.2)	0.872 *

* Mann–Whitney test; ** Independent Samples “t” test; *** χ2 test; GDM—gestational diabetes mellitus; NGT—normal glucose tolerance; HOMA-IR—homeostasis model assessment of insulin resistance; BMI—body mass index; FPG—fasting plasma glucose; FSI—fasting serum insulin; OGTT—oral glucose tolerance tests; OGTT-1h—1-h post-glucose load; OGTT-2h—2-h post-glucose load; HOMA-B—homeostasis model assessment of beta-cell function; HOMA-S—homeostasis model assessment for insulin sensitivity; QUICKI—quantitative insulin sensitivity check index; Glu/Zn—ratio between plasma glucose and serum zinc; Ins/Zn—ratio between insulin and zinc in serum; HOMA-IR/Zn—ratio between HOMA-IR and zinc.

**Table 2 ijms-25-12193-t002:** Comparison of ratios in GDM and NGT pregnant groups.

Parameters	GDM Pregnant Women (n = 54) Median (Interquartile Range)	NGT Pregnant Women; (n = 54) Median (Interquartile Range)	*p*-Value
Glu/Zn; mmol/L/ µmol/L	0.39(0.30–0.41)	0.29(0.24–0.35)	<0.001 *
Ins/Zn; µU/mL/ µmol/L	0.73(0.33–1.16)	0.48(0.25–0.70)	0.017 *
HOMA-IR/Zn	0.14(0.08–0.23)	0.06(0.03–0.09)	<0.001 *

* Mann–Whitney test; GDM—gestational diabetes mellitus; NGT—normal glucose tolerance; Glu/Zn—ratio between plasma glucose and serum zinc; Ins/Zn—ratio between insulin and zinc in serum; HOMA-IR/Zn—ratio between HOMA-IR and zinc.

**Table 3 ijms-25-12193-t003:** Correlations of Glu/Zn, HOMA-IR/Zn, Ins/Zn with insulin resistance and sensitivity in GDM pregnant group.

Indexes	HOMA-IR		HOMA-B		HOMA-S		QUICKI	
	Spearman’s rho	*p*-Value	Spearman’s rho	*p*-Value	Spearman’s rho	*p*-Value	Spearman’s rho	*p*-Value
Glu/Zn	0.212	0.123	0.196	0.156	−0.327	0.016	−0.252	0.066
HOMA-IR/Zn	0.937	0.001	0.691	0.001	−0.824	0.001	−0.895	0.001
Ins/Zn	0.799	0.001	0.698	0.001	−0.809	0.001	−0.879	0.001

Glu/Zn—ratio between plasma glucose and serum zinc; Ins/Zn—ratio between insulin and zinc in serum; HOMA-IR/Zn—ratio between HOMA-IR and zinc; HOMA-IR—homeostasis model assessment of insulin resistance; HOMA-B—homeostasis model assessment of beta-cell function; HOMA-S—homeostasis model assessment for insulin sensitivity; QUICKI—quantitative insulin sensitivity check index.

**Table 4 ijms-25-12193-t004:** Correlations of Glu/Zn, HOMA-IR/Zn, Ins/Zn with selected indexes of insulin resistance and sensitivity in NGT pregnant group.

Indexes	HOMA-IR		HOMA-B		HOMA-S		QUICKI	
	Spearman’s rho	*p*-Value	Spearman’s rho	*p*-Value	Spearman’s rho	*p*-Value	Spearman’s rho	*p*-Value
Glu/Zn	0.285	0.018	0.004	0.972	−0.196	0.107	−0.270	0.025
HOMA-IR/Zn	0.918	0.001	0.665	0.001	−0.846	0.001	−0.795	0.001
Ins/Zn	0.777	0.001	0.554	0.001	−0.724	0.001	−0.871	0.001

Glu/Zn—ratio between plasma glucose and serum zinc; Ins/Zn—ratio between insulin and zinc in serum; HOMA-IR/Zn—ratio between HOMA-IR and zinc; HOMA-IR—homeostasis model assessment of insulin resistance; HOMA-B—homeostasis model assessment of beta-cell function; HOMA-S—homeostasis model assessment for insulin sensitivity; QUICKI—quantitative insulin sensitivity check index.

## Data Availability

The data used to support the presented results of this survey are included in the article.

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
