# Peer review of "Association Between Zinc Status and Insulin Resistance/Sensitivity Check Indexes in Gestational Diabetes Mellitus"

_ijms, 2024, doi:10.3390/ijms252212193_

Round 1

Reviewer 1 Report

Comments and Suggestions for Authors

This is a review of ijms-3241733 “Association between Zinc Status and Insulin Resistance/ Sensitivity Check Indexes in Gestational Diabetes Mellitus” by Genova et al. The authors found the possibility of new diagnostic markers for gestational diabetes mellitus (GDM), such as Ins/Zn, Glu/Zn, and HOMA-IR/Zn. While there are no significant differences in plasma zinc concentration between GDM and normal glucose tolerance (NGT) pregnant women, GDM patients showed high levels of indexes which the authors tested compared to NGT pregnant group. They suggest that these new indexes could be a diagnosis of insulin resistance.

This reviewer did not find this study is not brand new and considered following points.

Major points:

1.         The authors showed only the data at second trimester and compared between GDM and NGT. There was no significant difference in Zn concentration among them, and there were significant differences in insulin, glucose and HOMA-IR. According to these data, Ins/Zn, Glu/Zn and HOMA-IR/Zn ratios are consistent with their single parameters. What is the difference between the ratio parameters and the single one? This led this reviewer considering the ratio parameters are not very useful. Why don’t the authors compare plasma zinc concentration before and after the onset of GDM, or before pregnancy and after the onset of GDM?

Minor points:

1.         In introduction, this reviewer considered that the authors should re-consider about contents of introduction. There are some sentences that seem unnecessary, for example L.49-52, L76-81, and etc.

2.         In table 3 and 4, the authors should omit the line which has no data (IR/Zn).

3.         In L.157-160, there seem to be contradiction between the manuscript and the data.

4.         As the authors mentioned, circadian rhythm affects the zinc concentration. However, the authors did not describe blood sample collection time. The authors should show this.

Comments on the Quality of English Language

Extensive editing of English language required.

Author Response

This is a review of ijms-3241733 “Association between Zinc Status and Insulin Resistance/ Sensitivity Check Indexes in Gestational Diabetes Mellitus” by Genova et al. The authors found the possibility of new diagnostic markers for gestational diabetes mellitus (GDM), such as Ins/Zn, Glu/Zn, and HOMA-IR/Zn. While there are no significant differences in plasma zinc concentration between GDM and normal glucose tolerance (NGT) pregnant women, GDM patients showed high levels of indexes which the authors tested compared to NGT pregnant group. They suggest that these new indexes could be a diagnosis of insulin resistance.

This reviewer did not find this study is not brand new and considered following points.

Major points:

  1. The authors showed only the data at second trimester and compared between GDM and NGT. There was no significant difference in Zn concentration among them, and there were significant differences in insulin, glucose and HOMA-IR. According to these data, Ins/Zn, Glu/Zn and HOMA-IR/Zn ratios are consistent with their single parameters. What is the difference between the ratio parameters and the single one? This led this reviewer considering the ratio parameters are not very useful. Why don’t the authors compare plasma zinc concentration before and after the onset of GDM, or before pregnancy and after the onset of GDM?

Dear Reviewer, Thank you for your valuable suggestion.  As the authors pointed in the manuscript, zinc is essential for life element with different biological functions in human body. During pregnancy the element is especially important because its link with insulin, with underlined metabolic changes, with fetus development and specific role of placenta. The authors’ idea is to load data for the second trimester when GDM is diagnosed and compared them to pregnant group with normal glucose tolerance because of specific response in zinc status and glucose metabolism during pregnancy as a specific physiological condition. The newly introduced ratios of Ins/Zn, HOMA-IR/Zn and Glu/Zn should be a subject for further research. The work is only initial attempt to suggest new surrogate biomarkers with potential for computational indexes in risk evaluation and diagnosis of insulin resistance during pregnancy. Despite the lack of statistical differences for the mentioned by the reviewer ratios, the authors’ point of view could present information for future perspectives in laboratory assessment for of insulin resistance and some specific metabolic alterations during pregnancy based on the link between zinc and insulin.

The authors are extremely grateful for the presented by the reviewer notes as they could serve as a point for further deeper clinical and laboratory observations on the discussed matter based even on possible nutritional strategy during pregnancy.

Minor points:

  1. In introduction, this reviewer considered that the authors should re-consider about contents of introduction. There are some sentences that seem unnecessary, for example L.49-52, L76-81, and etc.

Thanks for pointing out, it is corrected accordingly.

  1. In table 3 and 4, the authors should omit the line which has no data (IR/Zn).

Thank you for pointing out, it is corrected accordingly.

  1. In L.157-160, there seem to be contradiction between the manuscript and the data.

Thank you for pointing out, it is corrected accordingly.

  1. As the authors mentioned, circadian rhythm affects the zinc concentration. However, the authors did not describe blood sample collection time. The authors should show this.

Thank you for your valuable suggestion.

According to the recommendations for standardization, blood for all blood tests should be drawn in the morning (between 7 and 9 am) in a fasting state,12 h after the last meal. This requirement is important to limit the effect of postprandial response in every individual patient, to minimize the variation as a consequence of inter-individual heterogeneity and to control eventual circadian influence (Joint EFLM-COLABIOCLI Recommendation for venous blood sampling, Clin Chem Lab Med 2018; 56(12): 2015–2038). Diagnosis of GDM is made on the base of interpretation of OGTT (Oral Glucose Tolerance Test) results in accordance to the criteria of International Association of Diabetes and Pregnancy Study Groups and American Diabetes Association 2021. Guidelines of these organizations recommend 8-10 (no longer than 16 hours) fasting pause before blood collection for OGTT.

   The authors pointed in the section Materials and Methods (Sample collection and measurement) of the suggested manuscript that all the study participants are instructed to fast from 10 p.m. the night before as necessary step for standardization of pre-analytical phase in laboratory examinations thus keeping all recommendations mentioned above. 

Comments on the Quality of English Language

Extensive editing of English language required.

Thanks for pointing out; grammatical errors have been corrected throughout the manuscript.

Reviewer 2 Report

Comments and Suggestions for Authors

The paper titled "Association between Zinc Status and Insulin Resistance/ Sensitivity Check Indexes in Gestational Diabetes Mellitus" represents the role of one essential metals on incidence of gestational diabetes which sometimes could be fatal for mother and fetus. 

The paper is very well written with good conception and good results. Nevertheless I would like to pount some things which should be point out or correct. 

The introduction section should be shorten. Don't put notions that are already known (biochemistry book), but just new research data. 

Please don't say "abnormal" level line 104 but some proper word. 

Please explain more deeply what have you want to show with ROC analyses. 

The major issue is that you have 108 participants. How do you know that this number is rapresentative for good statistics?

Comments on the Quality of English Language

Please although the paper is written in good english there are some of correction to be done - as example - line 158 - did not reaches and more. line 349 - every one participant ???? all maybe.

Check  the english corrections with native english speaker who has nknowledge in medical science

Author Response

The paper titled "Association between Zinc Status and Insulin Resistance/ Sensitivity Check Indexes in Gestational Diabetes Mellitus" represents the role of one essential metals on incidence of gestational diabetes which sometimes could be fatal for mother and fetus. 

The paper is very well written with good conception and good results. Nevertheless I would like to pount some things which should be point out or correct. 

The introduction section should be shorten. Don't put notions that are already known (biochemistry book), but just new research data. 

Thank you for your valuable suggestion. The Introduction is shorten.

Please don't say "abnormal" level line 104 but some proper word. 

Thank you for pointing out, it is corrected accordingly.

Please explain more deeply what have you want to show with ROC analyses. 

Thank you for the question! ROC curves are used to define optimal cut-off points of numerical variables with certain sensitivity and specificity.

The major issue is that you have 108 participants. How do you know that this number is rapresentative for good statistics?

Thank you for your valuable suggestion! We calculated the sample for two means based on Mishu, F. A., Boral, N., Ferdous, N., Nahar, S., Sultana, G. S., Yesmin, M. S., Khan, N. Z. Estimation of serum zinc, copper and magnesium levels in bangladeshi women with gestational diabetes mellitus attending in a tertiary care hospital. Mymensingh Med J 2019, 28(1), 157–162. They provided information for mean zinc levels among GDM group and healthy controls: 43.93±5.48μg/dl and 67.30±7.81μg/dl respectively. We assumed that sd for both groups is 48μg/dl and the sample resulted in 46 in each arm with 95% confidence and 80% power. We added some more additional cases to compensate possible drop-outs.

Comments on the Quality of English Language

Please although the paper is written in good english there are some of correction to be done - as example - line 158 - did not reaches and more. line 349 - every one participant ???? all maybe.

Thanks for pointing out; grammatical errors have been corrected throughout the manuscript.

Check  the english corrections with native english speaker who has nknowledge in medical science

Thank you for your valuable suggestion. The manuscript is corrected accordingly.

Round 2

Reviewer 1 Report

Comments and Suggestions for Authors

This is a review of revised ijms-3241733 “Association between Zinc Status and Insulin Resistance/ Sensitivity Check Indexes in Gestational Diabetes Mellitus” by Genova et al.

This reviewer is still concerning the following points.

1.         The reviewer can understand that the authors suggest that new surrogate biomarkers in risk evaluation and diagnosis of insulin resistance during pregnancy. However, the authors tried to search for these markers from the GDM patients. This does not make sense to evaluate risks from these data, because they were already diagnosed as GDM. This reviewer thinks the single parameters should be enough to diagnose with GDM, because these parameters showed significant differences. The patients have already been diagnosed with GDM, why do you need new diagnostic markers for GDM?
For risk evaluation, the authors could assess the risks by comparing patients diagnosed with GDM with healthy controls using data prior to the second trimester of pregnancy which means before the patients have been diagnosed with GDM.

2.         In L.140-141, the authors mentioned there is a significance in NGT groups but not in GDM. However, there is a significance in GDM groups but not in NGT groups according to table 3 and 4.

3.         For blood sample collection time, the reviewer understands the author’s comment. If the authors follow the recommendations, the authors could describe the collection time “in the morning (between 7 – 9 am)”.

Author Response

  1. The reviewer can understand that the authors suggest that new surrogate biomarkers in risk evaluation and diagnosis of insulin resistance during pregnancy. However, the authors tried to search for these markers from the GDM patients. This does not make sense to evaluate risks from these data, because they were already diagnosed as GDM. This reviewer thinks the single parameters should be enough to diagnose with GDM, because these parameters showed significant differences. The patients have already been diagnosed with GDM, why do you need new diagnostic markers for GDM?
    For risk evaluation, the authors could assess the risks by comparing patients diagnosed with GDM with healthy controls using data prior to the second trimester of pregnancy which means before the patients have been diagnosed with GDM

Answer: The ambitious of the authors is to explore the link between zinc status and insulin resistance, evaluated by already known glucometabolic parameters. Diagnosed GDM is chosen as a clinical model for altered glycemic regulation. The same pregnancy, as a specific physiological condition, is characterized by complicated response to the status of essential trace elements copper and zinc. The clinical manifestation of zinc deficiency is variable, unspecific and depends on the degree and duration of zinc depletion. The increased demands for zinc in pregnancy could require additional supplementation. As the authors point in the manuscript, the present study is only one attempt to load data about the association about zinc and used in clinical practice indexes for insulin resistance in GDM. Computed calculations of the suggested ratios could be useful in the monitoring of the pregnancy course and for early detection of possible disorders of glucose metabolic control in relation to Zn as very important micronutrient. The suggested ratios are not definitive for the diagnosis but they are clinically beneficial for the pregnancy management especially in complicated situation of GDM.     

 In L.140-141, the authors mentioned there is a significance in NGT groups but not in GDM. However, there is a significance in GDM groups but not in NGT groups according to table 3 and 4

Answer: Yes, the authors are agree with this note - technical error, made in the text.

According to the Tables 3 and 4: In the manuscript should be put: There is weak correlation between ratio Glu/Zn and HOMA S with no significant difference in NGT group.

  1. For blood sample collection time, the reviewer understands the author’s comment. If the authors follow the recommendations, the authors could describe the collection time “in the morning (between 7 – 9 am)

Comment:

According to IFCC and EFLM recommendations, the blood for all test should be drawn 7-9 am.

For diabetes ADA recommend 8-10 fasting pause.

The authors put in the manuscript the following: The study participants were instructed to fast from 10 p.m. the night before- according ADA recommendations.

The authors are agreed to precise blood collection time, pointed in this sentence.

Round 3

Reviewer 1 Report

Comments and Suggestions for Authors

This is a review of revised ijms-3241733 “Association between Zinc Status and Insulin Resistance/ Sensitivity Check Indexes in Gestational Diabetes Mellitus” by Genova et al.

This reviewer eventually understands the suggestion by the authors.

Only one point, the authors should add this phrase “The suggested ratios are not definitive for the diagnosis but they are clinically beneficial for the pregnancy management especially in complicated situation of GDM.” into the Discussion or Conclusion part.

Author Response

Comment 1: This reviewer eventually understands the suggestion by the authors. Only one point, the authors should add this phrase “The suggested ratios are not definitive for the diagnosis but they are clinically beneficial for the pregnancy management especially in complicated situation of GDM.” into the Discussion or Conclusion part. 

Response 1: Dear Reviewer, thank you for this suggestion! The sentence is added at the end of the Conclusions. 
